# MAKE INTERVAL BOUND PROPAGATION GREAT AGAIN

## ABSTRACT

In various scenarios motivated by real life, such as medical data analysis, autonomous driving, and adversarial training, we are interested in robust deep networks. A network is robust when a relatively small perturbation of the input cannot lead to drastic changes in output (like change of class, etc.). This falls under the broader scope field of Neural Network Certification (NNC). Two crucial problems in NNC are of profound interest to the scientific community: how to calculate the robustness of a given pre-trained network and how to construct robust networks. The common approach to constructing robust networks is Interval Bound Propagation (IBP). This paper demonstrates that IBP is sub-optimal in the first case due to its susceptibility to the wrapping effect. Even for linear activation, IBP gives strongly sub-optimal bounds. Consequently, one should use strategies immune to the wrapping effect to obtain bounds close to optimal ones. We adapt two classical approaches dedicated to strict computations – Dubleton Arithmetic and Affine Arithmetic – to mitigate the wrapping effect in neural networks. These techniques yield precise results for networks with linear activation functions, thus resisting the wrapping effect. As a result, we achieve bounds significantly closer to the optimal level than IBPs.

## 1 INTRODUCTION

Deep neural networks find application in medical data analysis, autonomous driving, and adversarial training (Zhang et al., 2023) where safety-critical and robustness guarantees against adversarial examples (Biggio et al., 2013; Szegedy et al., 2014) are extremely important. The rapid development of artificial intelligence models does not correspond to their robustness (Luo et al., 2024). Therefore, certifiable robustness (Zhang et al., 2022; Ferrari et al., 2022) becomes an important task in deep learning. The aim of Neural Network Certification lies in rigorous validation of a classifier's robustness within a specified input region.

Most commonly applied certification method is interval bound propagation (IBP) (Gowal et al., 2018; Mirman et al., 2018). It is based on application of interval arithmetic, which allows to propagate the input intervals through a neural network. If in the case of classification tasks such propagation gives an unambiguous output then all elements of the interval inputs are guaranteed to have identical prediction. Therefore, we can control the behavior of predictions of a neural network in an explicit neighborhood of the input data. Among the approaches studied most extensively in robustness of neural networks is Certified Training (Singh et al., 2018; Mao et al., 2024). These certified training methods try to estimate and optimize the worst-case loss approximations of a network across an input domain defined by adversary specifications. They achieve this by computing an over approximation of the network reachable set through symbolic bound propagation techniques (Singh et al., 2019; Gowal et al., 2018). Interestingly, training techniques based on the least accurate bounds derived from interval-bound propagation (IBP) have delivered the best empirical performance (Shi et al., 2021; Mao et al., 2024).

The certification process uses the network's upper bound of the propagated input interval. Although classical IBP gives reasonable estimations in robust training, it ultimately fails in the certification of classically trained neural networks. In practice, for a given pre-trained networks, intervals that store intermediate values in a neural network evaluation increase exponentially with respect to number of layers, see Theorem 2.1. This phenomena is known as *the wrapping effect* (Neumaier, 1993), which in the context of neural networks applications was previously an unexplored area. Such a growth of obtained bounds makes them often dramatically sub-optimal in practical applications.

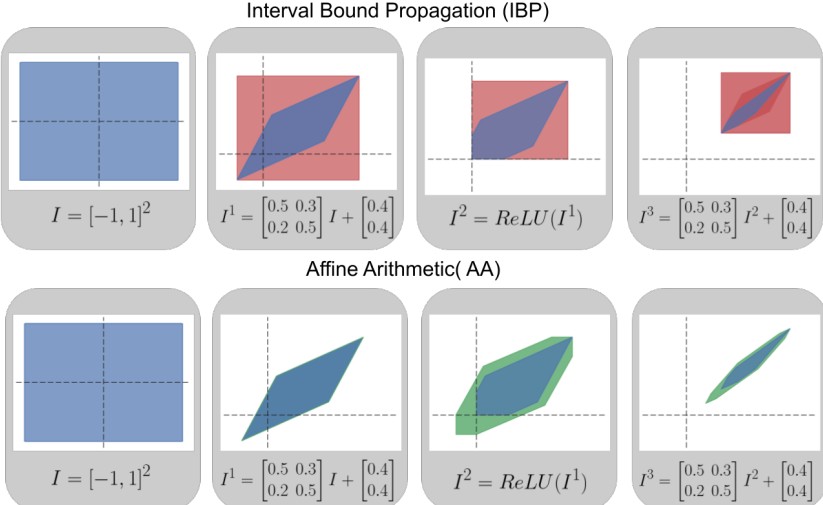

Figure 1: The figure presents how the interval is propagated throughout linear layers. By red color ■ we marked wrapping obtain by IBP and by green ■ by Affine Arithmetic. As we can see, Affine Arithmetic produces significantly lower wrapping effects. In the case of linear transformations, Affine Arithmetic gives an exact approximation. We can work with more complex objects than hyper-cubes from IBP and obtain bounds close to optimal ones. In Fig. 3 we present the procedure used in Affine Arithmetic to obtain $\text{ReLU}(I^1)$.

The aim of this paper is to analyse and adapt two existing methods for reduction of the wrapping effect to the context of neural networks applications: Doubleton Arithmetics (DA) (Mrozek & Zgliczyński, 2000) and Affine Arthmetic (AA) (de Figueiredo & Stolfi, 2004).

Doubletons is very special family of subsets of $\mathbb{R}^n$, which has been extensively used to reduce and control the wrapping effect in validated solvers to initial value problems of ODEs (Kapela et al., 2021). Direct application of interval arithmetics to propagate sets along trajectories, that is enclosing them into the Cartesian product of closed intervals, leads to accumulated overestimation known as the wrapping effect. This overestimation becomes larger and larger when we use smaller time steps $h$ of the underlying ODE solver. Lohner (Lohner, 1992) observed, that one can propagate coordinate system (approximate space derivative of the flow) between subsequent time steps along trajectories of the flow. This method proved to be very efficient and one of the reasons is that the for small time steps the mapping defined as a time shift along trajectories is close to identity. In this paper we adopt Doubleton Arithmetics to the context of neural networks. The main difference in comparison to ODEs is that the dimensions of subsequent layers in a network are usually different, while in ODEs we have a fixed dimension of the phase space. Moreover, in neural networks we often deal with non-smooth functions, such as ReLU. In Section 3 we will formally define doubleton representation and give algorithm for propagation of non-smooth functions in this arithmetics.

The second method, called Affine Arthmetic (AA) (de Figueiredo & Stolfi, 2004), is a special case of Taylor Models by (Berz & Makino, 1999; Makino & Berz, 2009). Here subsets of $\mathbb{R}^n$ are represented as a range of an affine map (often sparse) over a cube $[-1, 1]^m$, where $m$ in general is not related to $n$. Similarly to DA, evaluation of affine layers in AA causes no wrapping effect and it is sharp. In Section 3 and Appendix we show how to implement ReLU and softmax functions over a set represented in this way. The main numerical drawback of AA is that its computational cost is non-constant and depends on actual input arguments. Our experiments show that AA outperforms IBP in obtained bounds, see Fig. 1. Although AA and DA provide bounds of comparable sizes, AA is much faster (orders of magnitude) than DA on large networks. Thus, we recommend AA for evaluation of interval inputs through neural networks.

To make our approach completely certifiable, we need to have the full control over the numerical and rounding errors appearing in floating point arithmetics (Kahan, 1996). To obtain this we have decided to switch from Python-based networks to interval arithmetics (IEEE Std 1788.1-2017, 2018) in C++ with the use of the CAPD library (Kapela et al., 2021), which gives us certifiable control over the rounding errors. Consequently, to the best of the authors' knowledge, the presented approach is the first model that deals with rounding errors and obtains guaranteed boundaries.

Our contributions can be summarized as follows:

- We theoretically analyze the wrapping effect in the Neural Network certification task and show that classical IBP is sub-optimal even for linear transformations.
- We adapt two approaches, Doubleton Arithmetic and Affine Arithmetic, with full control over numerical and rounding errors to the neural network certification task.
- Using empirical evaluation, we show that Affine Arithmetic gives the best bounds of the neural network output and significantly outperforms classical IBP.

## 2  IBP AND WRAPPING EFFECT

In this section, we examine how interval bounds propagate through a linear layer. We show that the appearance of *wrapping effect*, even in the case of isometric transformations, leads to an exponential growth of interval bounds. Wrapping effect is typically studied in the context of strict estimations for solutions of dynamical systems, where the propagated set is at each iteration "wrapped" in the minimal interval bound (Neumaier, 1993). Therefore, applying the standard interval bound propagation layer after layer leads to an exponential increase of bounds.

Given a bounded set $X \subset \mathbb{R}^n$, by $\mathrm{IB}(X)$ (interval bounds) we denote the smallest interval bounding box for $X$. The aim of IBP (Interval Bound Propagation) lies in obtaining the IB for the processing of $X$ through a network, i.e. a series of possibly nonlinear maps. In the case of linear map $A = [a_{ij}]$, the optimal bounds are given by

$$\mathrm{IB}(A(x + [-r, r])) = Ax + [-|A|r, |A|r], \tag{1}$$

where $x + [-r, r] = \prod_i [x_i - r_i, x_i + r_i]$ and $|A| = [|a_{ij}|]$. To propagate an interval through the ReLU activation, we propagate the lower and upper bound separately: $\mathrm{ReLU}([x, y]) = [\mathrm{ReLU}(x), \mathrm{ReLU}(y)]$. We can propagate intervals through the standard network $\Phi$, which is represented as a sequence of mappings corresponding to the successive layers $y = \Phi(x) = \phi_k \circ \ldots \circ \phi_1(x)$. The aim of IBP is to obtain the estimate of $\mathrm{IB}(\Phi(x + [-r, r]))$, where commonly we restrict to the case when $r = \varepsilon \mathbb{1}$:

$$\mathrm{IB}(\Phi(x + \varepsilon[-\mathbb{1}, \mathbb{1}])), \text{ where } \mathbb{1} = (1, \ldots, 1) \in \mathbb{R}^n.$$

The standard classical approach used for IBP in the networks uses the naive iterative approach, where we process through each layer the interval bounds obtained from the previous one:

$$[I = x + [-r, r]] \rightarrow [I^1 = \mathrm{IB}(\phi_1(I))] \rightarrow [I^2 = \mathrm{IB}(\phi_2(I^1))] \rightarrow \ldots \rightarrow [y = \mathrm{IB}(\phi_k(I^k))].$$

Since we compute interval bound in each stage, the estimations are far from optimal; see Fig. 1. In practice, wrapping effects appear in neural networks. We will show that intervals grow exponentially, even for linear networks. We consider linear orthogonal ones, as they can be seen as the natural initialization of the deep network (Nowak et al.).

We will need the following lemma which proof is given in the Appendix.

**Lemma 2.1.** *Let $V = (V_1, \ldots, V_n)$ be a random vector uniformly chosen from the unit sphere in $\mathbb{R}^n$. Let $R$ be a random variable given by $R = |V_1| + \ldots + |V_n|$. Then*

$$\mathbb{E}(R) = \frac{\sqrt{2}}{\sqrt{\pi}} \sqrt{n} + O(1/\sqrt{n}), \ \mathbb{V}(R) = 1 + \frac{1}{\pi} + O(1/n).$$

Now we will show how a uniform interval bound is processed through an orthogonal map (isometry).

**Proposition 2.1.** *Let $U$ be a randomly chosen orthogonal map in $\mathbb{R}^n$. Then*

$$\mathrm{IB}(U([-\mathbb{1}, \mathbb{1}])) \approx \frac{\sqrt{2}}{\sqrt{\pi}} \sqrt{n} \cdot ([-\mathbb{1}, \mathbb{1}] + O(1/\sqrt{n})).$$

*Proof.* For each fixed $i = 1 \ldots, n$ the $i$-th row of $U$ is a random vector uniformly chosen from the unit sphere. Thus $U(x) = [U_1(x), \ldots, U_n(x)]^T$. By (1), $\mathrm{IB}(U_i([-\mathbb{1}, \mathbb{1}])) = [-R_i, R_i]$, where $R_i$ is a random variable given by $R_i = |U_{i1}| + \ldots + |U_{in}|$. Now by Lemma 2.1, $\mathbb{E}(R_i) = \frac{\sqrt{2}}{\sqrt{\pi}} \sqrt{n} + O(1/n)$, $\mathbb{V}(R_i) = 1 + \frac{1}{\pi} + O(1/n)$. By the Chebyshev inequality,

$$\mathrm{P}(|R_i - \mathbb{E}[R_i]| \geqslant a) \leqslant \frac{\mathbb{V}([R_i])}{a^2},$$

and consequently asymptotically for large $n$

$$\mathrm{P}(|R_i - \frac{\sqrt{2}}{\sqrt{\pi}}\sqrt{n}| \geqslant a) \leqslant \frac{1 + \frac{1}{\pi} + O(1/n)}{a^2}.$$

Consequently, with an arbitrary large probability

$$U_i \approx \frac{\sqrt{2}}{\sqrt{\pi}}\sqrt{n} \cdot [-1, 1] + O(1) = \frac{\sqrt{2}}{\sqrt{\pi}}\sqrt{n} \cdot \left([-1, 1] + O(1/\sqrt{n})\right).$$

$\square$

The following theorem shows that the standard IBP leads to an exponential increase of the bound with respect to the number of layers, even when the true optimal bound does not increase. We obtain the formal proof for the linear layers with orthogonal activations.

**Theorem 2.1.** *Let $U_1, \ldots, U_k$ be a sequence of orthogonal maps in $\mathbb{R}^n$, and let $U = U_k \circ \ldots \circ U_1$. Let $B_0 = [-\mathbb{1}, \mathbb{1}]$. Then*

$$\mathrm{IB}(U(B_0)) \approx \frac{\sqrt{2}}{\sqrt{\pi}}\sqrt{n}([-\mathbb{1}, \mathbb{1}] + O(1/\sqrt{n}).$$

*Let $B_i$ be defined iteratively by $B_i = \mathrm{IB}(U_i(B_{i-1}))$. Then*

$$B_k \approx (\frac{\sqrt{2}}{\sqrt{\pi}}\sqrt{n})^k([-\mathbb{1}, \mathbb{1}] + O(1/\sqrt{n}))$$

*Proof.* The proof follows from the recursive use of the previous proposition. $\square$

Observe, that the above theorem says, that applying standard interval bounds propagation layer after layer leads to exponential increase in the bound, as compared to the true optimal bound. This paper modifies two Doubleton and Affine Arithmetic models, which provide optimal bounds for linear transformations.

## 3 DOUBLETON AND AFFINE ARITHMETICS

As shown in the previous section, classical interval bound propagation leads to an exponential increase in the bounds, even for the case of most superficial linear networks, which implies that it is suboptimal for pre-trained networks. Consequently, we postulate that we should develop methods that obtain strict estimation in the case of linear networks. In this paper, we propose adapting two Doubleton and Affine Arithmetics models for deep neural networks.

**Doubleton Arithmetics** Doubleton is a class of subsets of $X \subset \mathbb{R}^n$ that are represented in the following form

$$X = \{x + Cr + Qq : r \in \mathbf{r}, q \in \mathbf{q}\},$$

for some $x \in \mathbb{R}^n$, $C \in \mathbb{R}^{n \times m}$, $Q \in \mathbb{R}^{n \times k}$ and $\mathbf{r} \subset \mathbb{R}^m$, $\mathbf{q} \subset \mathbb{R}^k$ are interval vectors (product of intervals) containing zero. For a possibly nonlinear function $f : \mathbb{R}^m \to \mathbb{R}^n$ and a compact set $W \subset \mathbb{R}^m$ we use doubletons to enclose range $f(W)$. The component $x + C\mathbf{r}$ is supposed to store linear approximation to $f$, while $Q\mathbf{q}$ stores accumulated errors (usually bounds on nonlinear terms) in certain (often orthogonal) coordinate system.

Such a family is one of the most frequently used in validated integration of ODEs (Lohner, 1992; Mrozek & Zgliczyński, 2000) and provides a good balance between accuracy (size of overestimation) add time complexity of operations on such objects. Here we would like to adopt it to the special case of neural networks. We have to extend doubleton arithmetics to functions with different dimensions of domain and codomain and also for non-smooth ReLU frequently used as an activate function in neural networks.

In the context of neural networks Doubleton Arithmetics is promising since we obtain sharp bound when mapping a doubleton by an affine transformation.

**Theorem 3.1.** *Evaluation of an affine function $A(t) = x_0 + Lt$ over a doubleton $X = x + C\mathbf{r} + Q\mathbf{q}$ is exact, that is*

$$A(X) = \left\{ \tilde{x} + \tilde{C}r + \tilde{Q}q : q \in \mathbf{q}, r \in \mathbf{r} \right\}, \text{ where } \tilde{x} = x_0 + Lx, \quad \tilde{C} = LC, \quad \tilde{Q} = LQ.$$

Consequently, we can process sets described by doubletons through linear layers without any wrapping effect. Enclosing a classical neural network activation function in Doubleton Arithmetics is more challenging. Below we proceed with the formulation how it can be done for a general nonlinear map.

Assume that $f : \mathbb{R}^n \to \mathbb{R}^d$ is a nonlinear function (even not continuous) and assume that for $z \in X = x + C\mathbf{r} + Q\mathbf{q}$ there holds

$$f(z) = x_0 + L(z - x) + e(z)$$

and let us assume that we have computed a bound $e(z) \in \mathbf{e}$ for $z \in X$. Then we have

$$
\begin{aligned}
f(z) &= x_0 + (LC)r + (LQ)q + e(z) = (x_0 + \text{mid}(\mathbf{e})) + (LC)r + (LQ)q + (e(z) - \text{mid}(\mathbf{e})) \\
&\in \tilde{x} + \tilde{C}\mathbf{r} + \tilde{Q}\tilde{\mathbf{q}},
\end{aligned}
$$

where $\tilde{x} = x_0 + \text{mid}(\mathbf{e}) \in \mathbb{R}^d$, $\tilde{C} = LC \in \mathbb{R}^{d \times m}$ and the term $\tilde{Q}\tilde{\mathbf{q}}$ is computed as follows. To simplify notation put $\Delta = \mathbf{e} - \text{mid}(\mathbf{e})$. Let $\tilde{Q} \in \mathbb{R}^{d \times n}$ and $A \in \mathbb{R}^{n \times d}$ be arbitrary matrices so that $\tilde{Q}A = \text{Id}_d$. Then for $q \in \mathbf{q}$ and $\delta \in \Delta$ we have

$$(LQ)q + \delta = \tilde{Q}A(LQq + \delta) = \tilde{Q}\left((ALQ)q + A\delta\right)$$

Now we define $\tilde{\mathbf{q}} := (ALQ)\mathbf{q} + A\Delta \subset \mathbb{R}^d$. It should be emphasized, that it is very important to first evaluate the product of matrices $(ALQ)$ and then multiply the result by the interval vector $\mathbf{q}$. Here is the place when we can reduce wrapping effect provided we make a *good choice* of $\tilde{Q}$ and $A$. There are various strategies for that.

**Strategy 1.** If $n = d$ and $LQ \in \mathbb{R}^{n \times n}$ is nonsingular then we may set $\tilde{Q} = LQ$ and $A = \tilde{Q}^{-1}$. Then $\tilde{\mathbf{q}} := \mathbf{q} + \tilde{Q}^{-1}\Delta \subset \mathbb{R}^n$.

**Strategy 2 - $QR$-decomposition.** If $d \geqslant n$ then we can first compute $QR$-decomposition of $LQ = \tilde{Q}R$, where $\tilde{Q} \in \mathbb{R}^{d \times d}$ is orthonormal. Then $A = \tilde{Q}^T$ and $R = ALQ \in \mathbb{R}^{d \times k}$. We can set $\tilde{\mathbf{q}} := R\mathbf{q} + \tilde{Q}^T\Delta \subset \mathbb{R}^d$. If $d < n$ then we may compute $QR$-decomposition of the leading $d \times d$ block of $LQ$ and proceed as before.

**Strategy 3 - $QR$-decomposition with pivots.** In **Strategy 2** we add a preconditioning step – that is permutation of columns of $LQ$ before $QR$-factorization. The permutation should take into account widths of components $\mathbf{q_i}$ and $\Delta_i$.

We can also try hybrid strategies in the case $n = d$. For instance, we can start from **Strategy 1** and if $LQ$ is singular or close to singular we switch to **Strategy 3**.

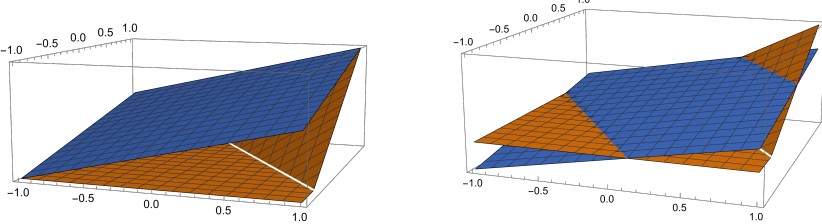

Figure 2: Graphs of $\text{ReLU}$ over a hyperplane crossing zero. (Left) first affine approximation of ReLU with $\tau = 1$, that is $\tilde{b}_0 + c \sum a_i t_i$ and (right) its final affine approximation $b_0 + c \sum a_i t_i$.

We can implement $\text{ReLU}$ and $\text{softmax}$ in Doubleton Arithmetics thanks to the above methodology. Consequently, we can track how the input interval is propagated through the neural network. Since the transformation by linear mapping does not cause the wrapping effect, we get much better estimates than the classical IBP.

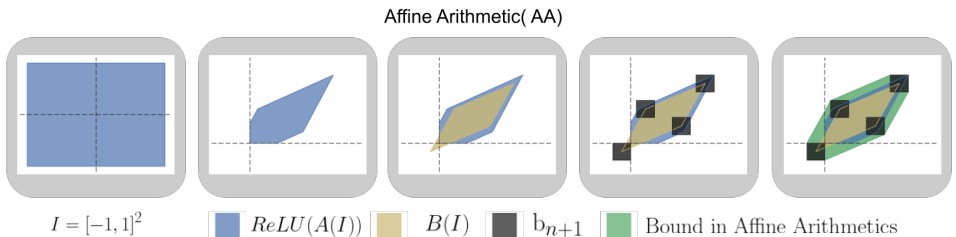

Figure 3: Affine Arithmetic works with more complicated shapes than hypercubes from IBP. In the example, we take Interval $I = [-1, 1]^2$ and see how AA produces an approximation of output from the linear layer with ReLu activation. We use affine transformation from Fig. 1. To approximate AA output from $\text{ReLU}(A(I))$, we first approximate nonlinear function $\text{ReLU}(A(\cdot))$ by linear $B(\cdot)$. Then, we propagate input interval $I$ through $B(\cdot)$. Then we add interval correction, which is equal to the maximal error between $\text{ReLU}(A(I))$ and $B(\cdot)$ denoted by $b_{n+1}$. Finally, we obtain bound in Affine Arithmetic in the case of mapping interval through a linear layer with linear activation.

**Affine Arithmetic**   The main drawback of the Doubleton Arithmetics is that it is expensive, because it involves multiplication of full dimensional non-sparse matrices. Affine arithmetics (de Figueiredo & Stolfi, 2004) is a concept of reducing overestimation in evaluating an expression in interval arithmetics coming from dependency, that is multiple occurrence of a variable in an expression.

Affine arithmetics keeps track of linear dependencies between variables through evaluation of an expression. Affine Arithmetic gives sharp bounds for linear transformations.

In affine aritmetics we represent subsets of $\mathbb{R}^n$ as a range of an affine functions $A([-1, 1]^m)$ for some affine map $A : \mathbb{R}^m \to \mathbb{R}^n$. Clearly the composition of $A$ with another affine map $B : \mathbb{R}^n \to \mathbb{R}^k$ is again an affine map $B \circ A : \mathbb{R}^m \to \mathbb{R}^k$ and thus the image of $[-1, 1]^m$ via $B \circ A$ is represented as an affine expression with no overestimation.

Let us present on an easy example the main property of affine arithmetics, which shows its superiority over interval arithemtics. Assume we have two expressions $A(x, y) = 1 + x + 2y$ and $B(x, y, z) = 1 - x - 2y + z$, where $x, y, z \in I := [-1, 1]$ and we would like to compute a bound on $A(I, I) + B(I, I, I)$. Evaluation in interval arithmetics gives

$$A(I, I) + B(I, I, I) \subset (1 + [-1, 1] + 2[-1, 1]) + (1 - [-1, 1] - 2[-1, 1] + [-1, 1]) = [-5, 9].$$

We see that multiple occurrence of a variable in an expression leads to large overestimation. In affine arithmetics we keep linear track of variables and only in the end we evaluate expression in interval arithmetics. This gives the following (sharp) bound

$$\begin{aligned} A(x, y) + B(x, y, z) &= 2 + z, \\ A(I, I) + B(I, I, I) &= 2 + [-1, 1] = [1, 3]. \end{aligned}$$

Because different affine functions may have different number of arguments it is convenient to treat them (formally) as functions $A : \ell^0 \to \mathbb{R}$, where $\ell^0$ is a set of sequences with all but finite number of non-zero elements. Then we have a straightforward interpretation of addition of such functions and multiplication of affine function by a scalar.

To adapt Affine Arithmetic to neural networks we need to implement $\text{ReLU}$ and $\text{softmax}$ functions. In the case of nonlinear transformation in Affine Arithmetic we approximate our nonlinear mapping by an affine function with known precision, see Fig. 2. Then we propagate our input interval throught this affine transformation and add a new interval equal to upper bound of the difference between linear approximation and original function, see Fig. 3.

In order to implement $\text{ReLU}$ in Affine Arithmetic let us consider an affine function

$$A(t_1, \cdots, t_n) = a_0 + \sum_{i=1}^{n} a_i t_i$$

defined on the cube $t = (t_1, \ldots, t_n) \in [-1, 1]^n =: I^n$ and assume $0 \in A(I^n)$. Clearly the composition $\text{ReLU} \circ A$ is nonlinear and the set $\text{ReLU}(A(I^n))$ cannot be represented exactly as a range of an affine function.

Our strategy is to find an affine map $B : \mathbb{R}^n \to \mathbb{R}$, which approximates well the composition $\mathrm{ReLU} \circ A$ on the hypercube $I^n$. Bound on the difference

$$\max_{t \in I^n} |\mathrm{ReLU}(A(t)) - B(t)|$$

will be treated as a new variable and finally the range $\mathrm{ReLU}(A(I^n))$ will be covered by a range of an affine function but with $n + 1$ variables. This scenario is visualised in Fig. 3.

We impose that $B$ is of the form

$$B(t) = b_0 + \sum_{i=1}^n (ca_i)t_i$$

for some $c \in \mathbb{R}$, that is $b_i = ca_i$ for $i > 0$. Put $S := \sum_{i=1}^n |a_i|$ and let $M := \sup_{t \in I^n} A(t) = a_0 + S$. Let $\tau \in [0, 1]$ be a parameter to be specified later. We impose that $B$ vanishes for its all arguments being $-1$, while it reaches maximum value in $I^n$ equal to $\tau M$ for all arguments equal to $1$ – see Fig.2 left panel. This gives the following system of equations with two unknowns

$$\widetilde{b}_0 - cS = 0, \qquad \widetilde{b}_0 + cS = \tau \cdot M.$$

The solution is $\widetilde{b}_0 = \frac{1}{2}\tau M$ and $c = \frac{1}{2}\tau M/S$. The graph of the first affine approximation $\widetilde{B}(t) = \widetilde{b}_0 + c\sum_{i=1}^n a_i t_i$ of $\mathrm{ReLU}(A(t))$ is shown in Fig. 2 left panel.

Now, we have to bound the difference between $\widetilde{B}$ and $\mathrm{ReLU} \circ A$ on $[-1, 1]^n$. By the choice of $c$ and $\widetilde{b}_0$ the maximal value of $\widetilde{B}(t)$ in the cube $[-1, 1]^n$ is $\tau M$. Hence

$$D_+ := \max_{t \in I^n} \left( \mathrm{ReLU}(A(t)) - \widetilde{B}(t) \right) = \max_{t \in I^n} \left( A(t) - \widetilde{B}(t) \right) = M - \tau M = M(1 - \tau). \quad (2)$$

The minimal value of this difference is achieved, when $(a_0 + \sum a_i t_i) = 0$, that is $\sum a_i t_i = -a_0$ – see Fig. 2 (left panel). This minimal value is then

$$D_- = \min_{t \in I^n} \left( \mathrm{ReLU}(A(t)) - \widetilde{B}(t) \right) = \min_{t \in I^n} \left( 0 - \widetilde{B}(t) \right) = -\widetilde{b}_0 + ca_0 = ca_0 - \frac{1}{2}\tau M. \quad (3)$$

Gathering (2)-(3) we obtain

$$\left( \mathrm{ReLU}(A(t)) - \widetilde{B}(t) \right) \in [D_-, D_+], \qquad \text{for } t \in [-1, 1]^n.$$

The above considerations lead to an algorithm for computation of ReLU in the affine arithmetics. Given coefficients $(a_0, \ldots, a_n)$ of an affine function $A(t_1, \ldots, t_n)$ we compute coefficients $(b_0, \ldots, b_{n+1})$ of an affine function $B(t_1, \ldots, t_{n+1})$ so that the range of $B(t)$, $t \in [-1, 1]^{n+1}$ covers the range of $\mathrm{ReLU}(A(t))$, $t \in [-1, 1]^n$ in the following way

$$S = \sum_{i=1}^n |a_i|, \quad M = a_0 + S, \quad c = \frac{1}{2}\tau M/S, \quad D_+ = M(1 - \tau), \quad D_- = ca_0 - \frac{1}{2}\tau M,$$

$$b_0 = \frac{1}{2}(\tau M + D_+ + D_-), \quad b_{n+1} = \frac{1}{2}(D_+ - D_-), \quad b_i = ca_i, \quad i = 1, \ldots, n.$$

There remains to explain how we choose the parameter $\tau \in [0, 1]$. Set $U = \max_{t \in [-1,1]^n} A(t) = a_0 + S$ and $L = \min_{t \in [-1,1]^n} A(t) = a_0 - S$. Experimentally we have found that the choice $\tau \approx \frac{U}{U-L} = \frac{a_0+S}{2S}$ gives reasonable small overestimation of $\mathrm{ReLU} \circ A$ and it is very fast to compute.

The computation of softmax in affine arithmetics is presented in Appendix.

## 4 EXPERIMENTS

In this section we present the results obtained by our two proposed methods: Affine and Doubleton Arithmetics. For a fixed point $x$ from the dataset, we define a box $B = x + \varepsilon[-\mathbb{1}, \mathbb{1}]$ and then we compare bounds on the output of a neural network $\Phi(B)$ obtained by means of IBP, DA and AA methods. Additionally we compute Lower Bound (LB) on $\Phi(B)$ as the smallest box (interval hull) containing the set $\{\Phi(\xi_k)\}_{k=1}^{1000}$, where $\{\xi_k\}_{k=1}^{1000} \subset B$ are randomly chosen points. All the experiments

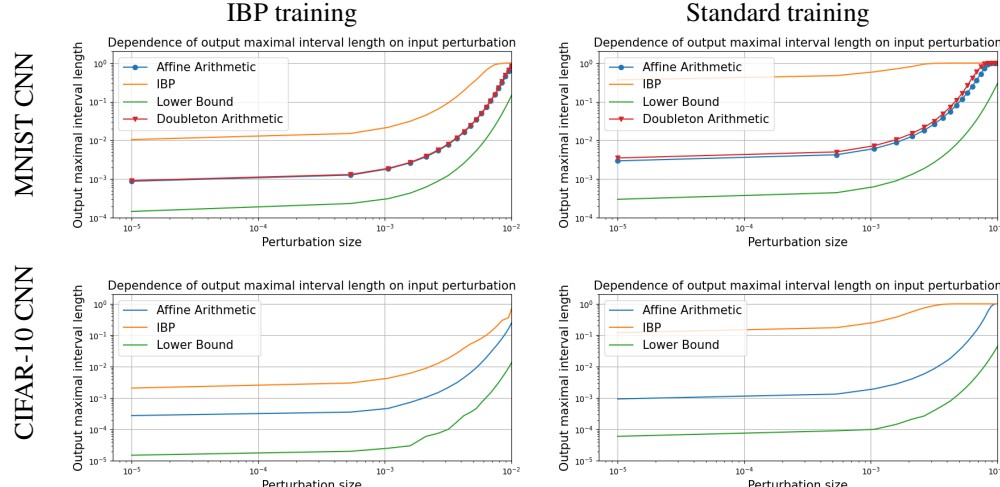

Figure 4: The average maximal diameter of the NN output measured for points near the classification boundary. The X axis represents the perturbation size applied to the data points, while the Y axis shows the average maximal diameter of the NN output in the logarithmic scale. As we can see, the AA and DA methods give better approximation of interval bounds than the IBP method. Note that the DA cannot be calculated for large CNN architectures according to CPU constraints. We can see that IBP training in relation to standard training allows to reduce wrapping effect.

are implemented in C++ with the full control over numerical and rounding errors obtained due to the use of CAPD library (Kapela et al., 2021).

The results presented for the MNIST, CIFAR-10, and SVHN datasets are shown only for the small CNN architecture unless stated otherwise. The results for the Digits dataset are presented using an MLP architecture. For more details about the training hyperparameters, architectures, datasets, and hardware used – see Section C in Appendix.

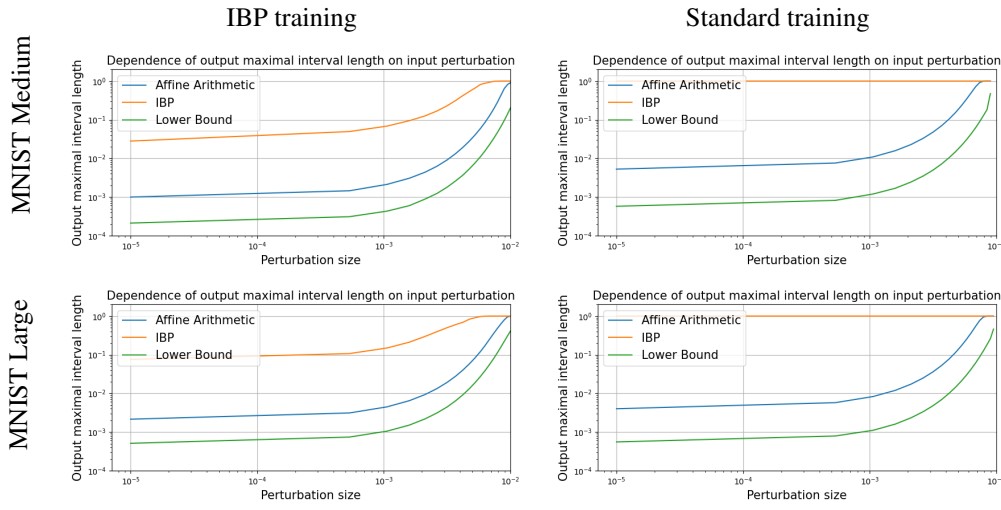

Figure 5: The average maximal diameter of the NN output measured for points near the classification boundary for the medium and large CNN architectures. The X axis represents the perturbation size applied to the data points, while the Y axis shows the average maximal diameter of the NN output in the logarithmic scale.

**Interval bounds for points sampled near the decision boundary**   We compare our methods on points sampled near the decision boundary. We start by selecting one point from each class. From these points, we sample one point and connect it to the remaining points with line segments. For each of these line segments, we sample a point that lies near the decision boundary. We then calculate the interval neural network output for each of these selected points and average maximal diameters of these intervals to assess the model's uncertainty. It is important to emphasize that these selected points may not be actual points from the real dataset, as they are, in fact, convex combinations of points from the real dataset. The experiments are conducted on the MNIST and CIFAR-10. We provide results for neural networks with weights obtained through a classical training procedure (without IBP training) and weights obtained through IBP training as well, for comparison.

As shown in Fig. 4, for a neural network trained using the IBP method ($\epsilon_{\text{train}} = 0.01$) and standard training, the AA and DA methods perform significantly better compared to the IBP method. It is important to highlight that both the AA and DA methods produce nearly identical results. We would like to emphasize, that the AA method gives useful answer even for large perturbation size $\epsilon$, while the IBP even for not very large perturbations gives useless outputs of length $1$, which means that the probability is somewhere between $0$ and $1$. Such phenomenon is well visible in each experiment we conducted – see Figs. 4, 5 and 6.

**Influence of network size on interval bounds**   It is a fair question how interval bounds change depending on the size of a neural network architecture. We address this question by using medium, and large CNN architectures for the MNIST dataset. The architectures were trained using the IBP method with $\epsilon_{\text{train}} = 0.01$, as well as without the IBP method for comparison.

For the medium and large CNN architectures trained on the MNIST dataset (Fig. 5), the AA method produces results close to those of the LB method, significantly outperforming the IBP method. These differences are particularly noticeable when IBP training is not applied. In this case, the AA method once again outperforms the IBP method, while the differences between the LB and AA methods are only slightly worse compared to the scenario when IBP training is used. It is also worth emphasizing that when medium and large CNN architectures are used without IBP training, the neural network becomes extremely uncertain about the investigated data points.

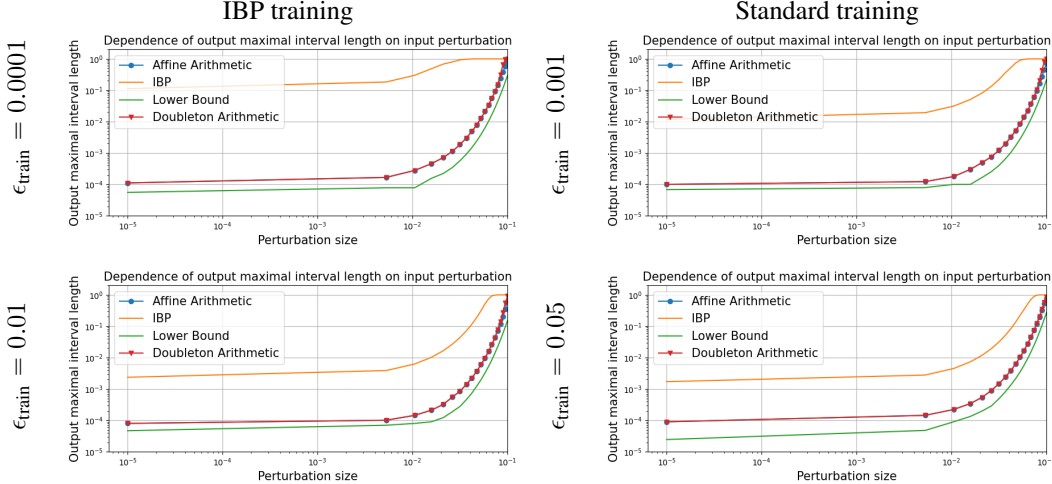

Figure 6: The average maximal diameter of the NN output measured for points near the classification boundary for network trained with different interval lengths $\epsilon_{\text{train}}$ for the Digits dataset. The X axis represents the perturbation size applied to the data points, while the Y axis shows the average maximal diameter of the NN output in the logarithmic scale.

**Influence of various perturbation sizes used in IBP-based training on the resulting interval bounds**   Generally, the presented plots in Fig. 6 show that the larger the perturbation size applied during IBP training, the smaller the difference between the results obtained using the IBP and AA/DA methods. However, it is important to emphasize that increasing the perturbation size during IBP

training makes training a neural network more difficult, leading to challenges in achieving satisfactory accuracy. Therefore, our proposed methods offer a much easier way to reduce the wrapping effect, and they can be applied regardless of whether IBP training is used, making them both more practical and efficient for real-world applications.

## 5 CONCLUSION

This paper analyzes *wrapping effect* in a neural network. We show that for linear models, interval bounds can grow exponentially. Such effects have a strong influence on the IBP certification of neural networks. To solve such a problem, we propose adapting two models from strict numerical calculations: Doubleton and Affine Arithmetics. Both models give sharp bounds for linear transformations. The experimental section shows that Affine Arithmetic returns bounds close to optimal within reasonable computational time.

**Limitations** Doubleton Arithmetics provides near-optimal bounds, but the computational complexity is $O(n^3)$, where $n$ is the largest dimension of the hidden layers. Even for the small CNN architecture on the CIFAR-10 and SVHN datasets, the computation time was unacceptably high.

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

## A  INTEGRAL COMPUTATIONS

We will show the following lemma, which gives a detailed estimations for Lemma 2.1.

**Lemma A.1.** *Let $V = (V_1, \ldots, V_n)$ be a random vector uniformly chosen from the unit sphere in $\mathbb{R}^n$. Let $R$ be a random variable given by*

$$R = |V_1| + \ldots + |V_n|.$$

*Then*

$$ER = \frac{2n}{n-1}\frac{\Gamma(\frac{n}{2})}{\sqrt{\pi}\Gamma(\frac{n-1}{2})}, \; ER^2 = 1 + \frac{2}{\pi}(n - \frac{1}{n-2}), \; VR = ER^2 - (ER)^2.$$

*Moreover, we have the asymptotics*

$$ER = \frac{\sqrt{2}}{\sqrt{\pi}}\sqrt{n} + \frac{1}{2\sqrt{2\pi n}} + O(n^{-3/2}), \; VR = 1 + \frac{1}{\pi} + O(1/n).$$

*Proof.* To calculate $ER$, we compute

$$ER = \frac{1}{S_{n-1}}\int_{x:\|x\|=1} |x|_1 dS(x) = \frac{1}{S_{n-1}}\int_{x:\|x\|=1} x_1 + \ldots + x_n dS(x) =$$

$$\frac{2^n}{S_{n-1}}\int_{x:\|x\|=1,x_1,\ldots,x_n\geqslant 0} x_1+\ldots+x_n dS(x) = \frac{n2^n}{S_{n-1}}\int_{x:\|x\|=1,x_1,\ldots,x_n\geqslant 0} x_1 dS(x) =$$

$$\frac{2n}{S_{n-1}}\int_{x:\|x\|=1,x_1\geqslant 0} x_1 dS(x) = \frac{2n}{S_{n-1}}\int_0^1 x_1 S_{n-2}(\sqrt{1-x_1^2})^{n-2}\cdot\frac{1}{\sqrt{1-x_1^2}}dx_1 =$$

$$\frac{2nS_{n-2}}{S_{n-1}}\int_0^1 x_1(1-x_1^2)^{\frac{n-3}{2}}dx_1 = \frac{nS_{n-2}}{S_{n-1}}\int_0^1 t^{\frac{n-3}{2}}dt =$$

$$=\frac{2nS_{n-2}}{(n-1)S_{n-1}} = \frac{2n}{n-1}\frac{\Gamma(\frac{n}{2})}{\sqrt{\pi}\Gamma(\frac{n-1}{2})}$$

Finally we obtain the assymptotic expansion

$$\approx\frac{2}{\sqrt{\pi}}[\frac{\sqrt{n}}{\sqrt{2}} + \frac{1}{4\sqrt{2n}} + O(1/n^{3/2})]. = \frac{\sqrt{2}}{\sqrt{\pi}}\sqrt{n} + \frac{1}{2\sqrt{2\pi n}} + O(n^{-3/2}).$$

We proceed to computation of $R^2$. We have

$$ER^2 = \frac{1}{S_{n-1}}\int_{x:\|x\|=1}(|x|_1+\ldots+|x_n|)^2 dS(x)$$

$$=\frac{1}{S_{n-1}}\int_{x:\|x\|=1}|x|_1^2+\ldots+|x_n|^2 dS(x) + \frac{n(n-1)}{S_{n-1}}\int_{x:\|x\|=1}|x|_1|x_2|dS(x) =$$

$$=1 + \frac{4n(n-1)}{S_{n-1}}\int_{x:\|x\|=1,x_1,x_2\geqslant 0} x_1 x_2 dS(x)$$

Now

$$\int_{x:\|x\|=1,x_1,x_2\geqslant 0} x_1 x_2 dS(x) = \int_{x_1,x_2\geqslant 0,x_1^2+x_2^2\leqslant 1} x_1 x_2 S_{n-3}\sqrt{1-(x_1^2+x_2^2)}^{n-3}\frac{1}{\sqrt{1-(x_1^2+x_2^2)}}dx_1 dx_2$$

Now

$$\int_{x_1,x_2\geqslant 0,x_1^2+x_2^2\leqslant 1} x_1 x_2(1-(x_1^2+x_2^2))^{n/2-2}dx_1 dx_2 = \int_0^{\pi/2}\int_0^1 r\sin\phi r\cos\phi(1-r^2)^{n/2-2}rdrd\phi$$

$$=\int_0^{\pi/2}\frac{\sin 2\phi}{2}d\phi\cdot\frac{1}{2}\int_0^1(1-t)t^{n/2-2}dt = \frac{1}{4}\cdot(\frac{2}{n-2} - \frac{2}{n}) = \frac{1}{n(n-2)}.$$

Finally

$$ER^2 = 1 + 4\frac{n-1}{n-2}\frac{S_{n-3}}{S_{n-1}} = 1 + 4\frac{n-1}{n-2}\frac{n/2}{\pi} = 1 + \frac{2}{\pi}(n - \frac{1}{n-2}).$$

Clearly $VR = ER^2 - (ER)^2$, which trivially yields the asymptotic expansion

$$VR = ER^2 - (ER)^2 = 1 + \frac{1}{\pi} + O(1/n).$$

$\square$

## B  DOUBLETON AND AFFINE ARITHMETICS

**Softmax in affine arithmetics**  Assume we have an affine function $At = x + Lt$ defined on the cube $t = (t_1, \ldots, t_n) \in [-1, 1]^n =: I^n$, with $x \in \mathbb{R}^m$, $L \in \mathbb{R}^{m \times n}$. Our goal is to find an affine map

$$Bt = \tilde{x} + \tilde{L}t$$

and a vector $e \in \mathbb{R}^m$, so that for $i = 1, \ldots, m$ there holds

$$\max_{t \in I^n} |(\text{softmax}(A(t)) - B(t))_i| \leqslant e_i.$$

The vector $\tilde{x}$ and the matrix $\tilde{L}$ will be computed from first order Taylor expansion of softmax. A bound on error term $e$ will be computed from second derivatives.

Recall, that for $z \in \mathbb{R}^m$

$$\text{softmax}(z) = (s_1, \ldots, s_m) := \left( \frac{\exp(z_i)}{\sum_{j=1}^m \exp(z_j)}, \ldots, \frac{\exp(z_m)}{\sum_{j=1}^m \exp(z_j)} \right).$$

In order to avoid numerical instabilities in evaluation of the above expression we take $R = \|z\|_\infty$ and compute $\text{softmax}(z) = \text{softmax}(z_1 - R, \ldots, z_m - R)$.

It is well known that the Jacobian of softmax is given by

$$J(z) := D\text{softmax}(z) = \begin{bmatrix} s_1(1 - s_1) & -s_1 s_2 & \ldots & -s_1 s_m \\ -s_1 s_2 & s_2(1 - s_2) & \ldots & -s_2 s_m \\ \vdots & \vdots & \ddots & \vdots \\ -s_m s_1 & \ldots & -s_{m-1} s_m & s_m(1 - s_m) \end{bmatrix}.$$

Thus, we can compute a linear approximation of softmax by

$$B(t) = \tilde{x} + \tilde{L}t = \text{softmax}(x) + (J(x)L)\,t.$$

In the above $\text{softmax}(x)$ and $J(x)$ are evaluated at a single point and therefore neither dependency error nor wrapping effect is present.

The error term $e_i$ can be bounded using second order Taylor expansion. We would like to find a bound

$$\left| (I^n)^T D^2 g_i(I^n) I^n \right| \leqslant e_i, \quad i = 1, \ldots, m,$$

where $g(t) = \text{softmax}(x + Lt)$. Differentiation of $Dg(t) = J(x + Lt)L$ gives

$$D^2 g_i(t) \quad = \quad L^T DJ_i(x + Lt)L.$$

There remains to derive formula for $DJ_i(z)$. Differentiation gives

$$\frac{\partial J_{ij}(z)}{\partial z_c} \quad = \quad \frac{\partial}{\partial z_c}(\delta_{ij} s_i - s_i s_j)$$
$$= \quad (\delta_{ijc} s_i - \delta_{ij} s_i s_c) - (\delta_{ic} s_i - s_i s_c)s_j - s_i(\delta_{jc} s_j - s_j s_c)$$
$$= \quad \delta_{ijc} s_i - \delta_{ij} s_i s_c - \delta_{ic} s_i s_j - \delta_{jc} s_i s_j + 2 s_i s_j s_c.$$

Evaluation of the above formula in interval arithmetics leads to a rough bound on the error term $e_i$. We will show, however, that increasing time complexity we can significantly reduce dependency problem in this expression.

**Evaluation of $g_i$ and products $g_i g_c$ and $g_i g_c g_r$.**  We have

$$g_i(t) = \frac{\exp(x_i + L_{i1} t_1 + \ldots + L_{in} t_n)}{\sum_{j=1}^m \exp(x_j + L_{j1} t_1 + \ldots L_{jn} t_n)} \tag{4}$$

Dependency in (4) can be reduced using equivalent formula

$$g_i(t) = \frac{\exp(y_i)}{\sum_{j=1}^m \exp(y_j + (L_{j1} - L_{i1})t_1 + \ldots + (L_{jn} - L_{in})t_n)}, \tag{5}$$

where $R = \max_{i=1,\ldots,m} x_i$ and $y_i = x_i - R$.

Let us recall an important in this context property of interval arithmetics. It is well known that multiplication is not distributive, that is for intervals $a, b, c$ there holds $a(b + c) \subset ab + ac$. However, if all intervals are nonnegative then we have equality. Such situation appears in evaluation of the product

$$
\begin{aligned}
g_i(t)g_c(t) &= \left(\frac{\exp(y_i)}{\sum_{j=1}^m \exp\left(y_j + \sum_{p=1}^n (L_{jp} - L_{ip})t_p\right)}\right)\left(\frac{\exp(y_c)}{\sum_{k=1}^m \exp\left(y_k + \sum_{p=1}^n (L_{kp} - L_{cp})t_p\right)}\right) \\
&= \frac{\exp(y_i + y_c)}{\sum_{j,k=1}^m \exp\left(y_j + y_k + \sum_{p=1}^n (L_{jp} + L_{kp} - L_{ip} - L_{cp})t_p\right)}.
\end{aligned}
$$

The above two expressions, when evaluated in interval arithmetics, may lead to different bounds (of course they can be intersected). Time complexity of the second evaluation is $O(M^2N)$ while direct evaluation (first expression) is of order $O(MN)$. However, $\mathrm{softmax}$ is applied to the output of last layer in a neural network, which is usually of low dimension and therefore this should not be a serious additional cost.

Similarly, we can evaluate the product of three functions as

$$
g_i(t)g_c(t)g_r(t) = \frac{\exp(y_i + y_c + y_r)}{\sum_{j,k,s=1}^m \exp\left(y_j + y_k + y_s + \sum_{p=1}^n (L_{jp} + L_{kp} + L_{sp} - L_{ip} - L_{cp} - L_{rp})t_p\right)}
$$

and intersect the result with direct multiplication of three intervals $g_i(t)$, $g_c(t)$ and $g_r(t)$.

## C    EXPERIMENTAL SETTING

**Datasets**    We use the following publicly available datasets: 1) MNIST dataset, consisting of 60,000 training and 10,000 testing $28 \times 28$ pixel gray-scale images of 10 classes of digits; 2) CIFAR-10 dataset, consisting of 50,000 training and 10,000 testing $32 \times 32$ colour images in 10 classes; 3) SVHN dataset, consisting of 600,000 $32 \times 32$ pixel colour images of 10 classes of digits; 4) Digits dataset, consisting of 1797 $8 \times 8$ pixel gray-scale images of 10 classes of digits.

**Architectures**    We use three CNN architectures (small, medium and large) as defined in Table 1 in Gowal et al. (2018). Additionally, we consider an MLP architecture consisting of four hidden layers with 100 neurons per layer. A classification head is added on top of these layers.

**Training parameters**    During training, we use the Adam optimizer Kingma & Ba (2017) with the default configuration of $\beta_1 = 0.9$ and $\beta_2 = 0.999$, but with different learning rates ($lr$) across all datasets. We consistently use the ReLU activation function. Whenever a scheduler is mentioned, we apply the MultiStepLR scheduler with a default multiplicative learning rate decay factor set to 0.1. The scheduler steps are applied twice: once after $\frac{1}{3}$ of the total number of iterations and once after $\frac{2}{3}$ of the total number of iterations. Additionally, there is a parameter $\kappa$ scheduled over the entire training process as $\kappa_i = \max\{1 - 0.00005 \cdot i, \kappa_{max}\}$, where $i$ denotes the current training iteration and $\kappa_{max}$ is set to 0.5. A perturbation value $\epsilon$ grows linearly from 0 at the beginning of training to the $\epsilon_{max}$ hyperparameter value at the midpoint of the total number of iterations. The considered $\epsilon_{max}$ values are from the set $\{0.0001, 0.001, 0.01, 0.05, 0.1\}$ and remain the same regardless of the architecture used. We use 10% of the training samples as the validation set.

- For the MNIST, SVHN, and CIFAR-10 datasets, we train small, medium, and large CNNs using the best set of hyperparameters identified in Gowal et al. (2018). We apply the same normalization and augmentation scheme. The only differences are in the epsilons $\epsilon$ used during training and the number of epochs. We decreased the total number of epochs for the CIFAR-10 and SVHN datasets to 100 for the large CNN.

- For Digits, we train the MLP for 50 epochs with batch sizes of 32. No normalization or augmentation is applied. The rest of the hyperparameters remain the same as for the MNIST, SVHN, and CIFAR-10 datasets.

**Hardware and software resources used** The implementation is done in Python 3.10.13, utilizing libraries such as PyTorch 2.3.1 with CUDA support, NumPy 1.26.4, Pandas 2.1.1, and others. Most computations are performed on an NVIDIA GeForce RTX 4090 GPU, with some training sessions also conducted on NVIDIA GeForce RTX 3080 and NVIDIA DGX GPUs. The experiments involving Affine and Doubleton Arithmetics were implemented using the CAPD library (Kapela et al., 2021).

# D EXPERIMENTAL RESULTS

**Interval bounds for partially masked data** In this subsection, we aim to present the interval bounds obtained through a neural network for data from the Digits and SVHN datasets. We sample 10 points, each belonging to a single class, and apply a mask where 50% of the values are masked (replaced by zero) and the remaining values stay unchanged. We then measure the average diameter of the neural network output.

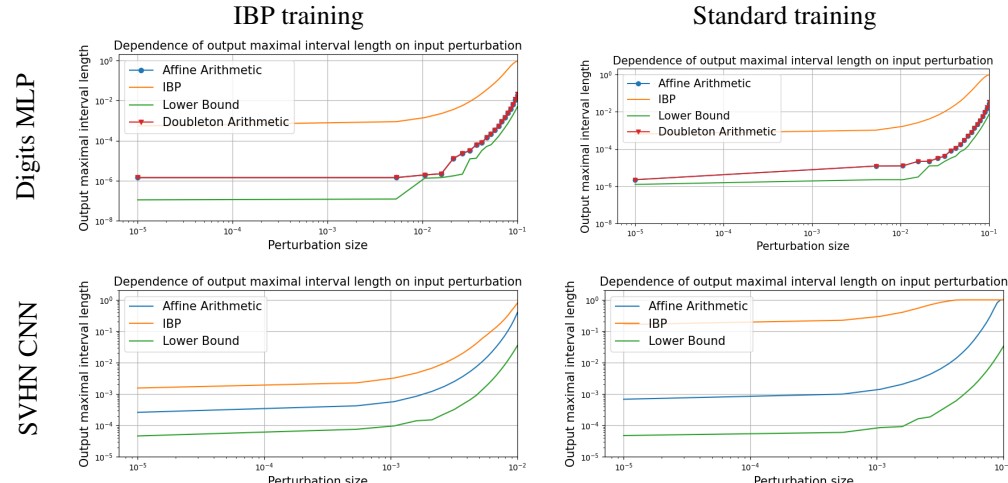

Figure 7: The average maximal diameter of the NN output measured for points near the classification boundary in the case where parts of the images was masked. The X axis represents the perturbation size applied to the data points, while the Y axis shows the average maximal diameter of the NN output in the logarithmic scale.

Even for partially masked data, the output interval bounds obtained using the AA and DA methods are very close to those of the LB methods (Fig. 7), significantly outperforming the IBP method.

These results indicate that the AA and DA methods, compared to the IBP method, effectively minimizes the wrapping effect in neural networks. Consequently, these methods can be regarded as a viable approach for quantifying the uncertainty of a neural network's output when some pixels of an image are masked with a value of 0.

