# OpenReview forum: "Make Interval Bound Propagation great again"
_ICLR.cc/2025/Conference — ICLR 2025 Conference Withdrawn Submission_

### Official Review · Reviewer_HaRJ · 2024-10-24

**Soundness:** 3
**Presentation:** 2
**Contribution:** 1
**Rating:** 3
**Confidence:** 5

**Summary:**

The paper explores the performance of bound propagation methods in the context of neural network certification. The paper demonstrates theoretically and empirically that bounds obtained by IBP (Interval Bound Propagation) produce exponentially growing over-approximations due to the wrapping effect. The authors propose two alternatives to IBP, namely Dubleton Arithmetic (DA) and Affine Arithmetic (AA) that produce tighter and closer to optimal over-approximations when compared to IBP.

**Strengths:**

The paper is well-structured and easy to read. The proofs are (mostly) easy to follow.

Using bounding methods from the field of ODE solvers in Neural Network Certification is an interesting direction.

The experiments are well organized and the results are presented and interpreted in insightful ways.

**Weaknesses:**

**Lack of discussion on relevant related work**

* The first claimed contribution of the paper (theoretical analysis of the wrapping effect in the case of IBP bounds) has been widely discussed in multiple recent works: Shi et al. (2021)[1] show that IBP bounds grow by a factor of $O(\sqrt n)$ after each affine layer, leading to exponential growth for deep networks. Various works [2,3,4,5,6] have proposed methods to compute tighter bounds for certification because IBP was not good enough. Other works [7,8,9] also acknowledge and experimentally verify the concept of exploding IBP bounds in the setting of Certified Training. Consequently, none of the theoretical results presented in section 2 of the paper can be considered novel.

* The Affine Arithmetic bound propagation method has been extensively used for Neural Network Certification in the literature under different names: Zonotope Abstraction [2], Symbolic Linear Relaxation [3], Fast-Lin [4], none of which have been discussed in the paper. Moreover, [2] is wrongly cited in Introduction (line 041) as a Certified Training method, when it is in fact a Certification method.

* Other works [5,6] focus on propagating linear constraints through the network. Most SOTA certification methods [10,11] combine this with further optimization such as multi-neuron analysis and branch-and-bound for analyzing unstable neurons.

* For the claim that their proposed methods are novel and better than existing methods, the authors should include discussions and justifications about the novelty and significance of their work when compared to previous work.

**Lack of experimental comparisons to related work**

* The authors only compare the results obtained by AA and DA with results obtained by IBP and with the Lower Bound obtained by sampling random points, but provide no comparisons with other existing certification methods. I recommend including some figures or tables with such comparisons.

* The paper presents plots of the maximal interval length for different networks and methods, but there are no results concerning the actual certification power of the methods (e.g. number of certified samples). I recommend including some figures or tables with such evaluations.

**Minor points**

* The notation mid() is not defined. Although a reader might infer from context that it refers to the middle point of the interval, it would be better to offer a formal definition or at least a reference to its definition from [12].

* Line 234: It is not clear if the $n$ considered for the dimensions of $\tilde Q$ and $A$ is the same as the dimension of the input space. In strategy 2 the matrices $\tilde Q$ and $A$ have dimensions $d \times d$ instead of $n \times d$ even if $d \neq n$.

* Figure 6 contains IBP training and Standard Training as titles for the two columns, but from the caption I understand there is only IBP training involved under different perturbation sizes.

* It would be nice to have a table with network architectures and hyperparameters used in the appendix instead of just referencing prior work.

**References**

[1] Shi et al., Fast certified robust training with short warmup, NeurIPS 2021

[2] Singh et al., Fast and effective robustness certification, NeurIPS 2018

[3] Wang et al., Efficient Formal Safety Analysis of Neural Networks, NeurIPS 2018

[4] Weng et al., Towards fast computation of certified robustness for relu networks, ICML 2018

[5] Zhang et al., Efficient neural network robustness certification with general activation functions, NeurIPS 2018

[6] Singh et al., An abstract domain for certifying neural networks, POPL 2019

[7] Muller et al., Certified training: Small boxes are all you need, ICLR 2023

[8] De Palma et al., Expressive losses for verified robustness via convex combinations, ICLR 2024

[9] Mao et al., Understanding certified training with interval bound propagation, ICLR 2024

[10] Wang et al., Beta-CROWN: Efficient Bound Propagation with Per-neuron Split Constraints for Neural Network Robustness Verification, NeurIPS 2021

[11] Ferrari et al., Complete Verification via Multi-Neuron Relaxation Guided Branch-and-Bound, ICLR 2022

[12] Mrozek and Zgliczyński, Set arithmetic and the enclosing problem in dynamics, Annales Polonici Mathematici 2000

**Questions:**

1. In the definition of DA, why is the term $C\textbf{r}$ needed? Why can’t this term just be combined with $Q\textbf{q}$?
2. I would like to see a comparison between the proposed methods and different SOTA certification methods regarding certification power and time (both convex-relaxation-based and multi-neuron or branch-and-bound-based ones).
3. A better lower bound might be obtained by using adversarial samples (i.e. generated by PGD adversarial attacks) instead of just randomly sampled points. Could the authors provide some comparisons in this regard?
4. Is there any significant improvement gained from using the CAPD C++ library for having full control over numerical and rounding errors? Without a proper discussion and/or experimental results, the computational overhead seems hard to justify.

---

### Official Review · Reviewer_L9KJ · 2024-11-03

**Soundness:** 3
**Presentation:** 2
**Contribution:** 2
**Rating:** 3
**Confidence:** 3

**Summary:**

This paper first studies the wrapping effect of IBP, which causes large approximation errors in forward propagation. The authors then propose to apply two existing methods, Doubletons and Affine Arithmetic to mitigate the wrapping effect, inspired by a similar problem in ODE. Experiments show that the proposed method can achieve a much tighter bounds compared to original IBP.

**Strengths:**

1. Considering the approximation error in IBP from a different perspective is very interesting to me.
2. The experiment results are strong compared to the original IBP method.

**Weaknesses:**

1. The conclusions about IBP wrapping effect has been discussed in many existing works such as [1], with a very similar theoretical result.
2. The proposed method looks very similar to the forward CROWN [2], which uses affine functions to bound the neural networks.

Therefore, the novelty of this paper is questionable given these relevant existing works.



[1] Fast certified robust training with short warmup.
[2] Automatic Perturbation Analysis for Scalable Certified Robustness and Beyond.

**Questions:**

Please see the weaknesses above.

---

### Official Review · Reviewer_wwup · 2024-11-04

**Soundness:** 3
**Presentation:** 2
**Contribution:** 3
**Rating:** 5
**Confidence:** 3

**Summary:**

The paper suggests a novel method, Doubleton and Affine Arithmetic, to control rounding errors in NN certification tasks by removing the wrapping effect in IBP. The theoretical findings are supported by experimental evidence.

**Strengths:**

1. Certification of pre-trained NNs is an important task for many different applications. The paper suggests novel theoretical approaches to achieve tighter bounds.

2. The intuition is well-explained, and the experiments are well-organized and thoroughly analyzed.

**Weaknesses:**

1. The method is only applicable for the linear transformation. However, there are various practical models with non-linear activation functions, which make its application is limited.

2. The time-complexity is too high for real applications such as ResNet and transformer-based models.

**Questions:**

Do you have any suggestions for practical use?

---

### Official Review · Reviewer_gjVZ · 2024-11-04

**Soundness:** 2
**Presentation:** 3
**Contribution:** 2
**Rating:** 3
**Confidence:** 4

**Summary:**

This study investigates methods to enhance the precision of Interval Bound Propagation (IBP) in the context of neural network certification, specifically under $\ell_\infty$-perturbations. The research demonstrates IBP's vulnerability to the wrapping effect and proposes the implementation of Dubleton Arithmetic and Affine Arithmetic as mitigation strategies.

**Strengths:**

1. Precisely certifying the robustness of neural networks remains a well-motivated problem.

2. This paper is well-written.

**Weaknesses:**

1. Overall, the setting is ad-hoc. The motivation of this work is the wrapping effect, specifically, the precision loss for a sequence randomly orthogonal maps is exponential. This is not a realistic setting. Moreover, the interpretation of Proposition 2.1 in the context of IBP is questionable.

2. Novelty is limited. The mitigation strategies, dubleton arithmetic and affine arithmetic, were already known. This work adapts these two known techniques to the problem.

3. The baseline is weak. The empirical evaluation only employs IBP as a baseline while overlooking state-of-the-art CROWN variants for $\ell_\infty$ certification.

**Questions:**

Could the authors give a clarification of proposition 2.1 in the context of IBP?

---

### Note · Authors · 2024-11-21

**Comment:**

We will correct the paper.

**Withdrawal Confirmation:**

I have read and agree with the venue's withdrawal policy on behalf of myself and my co-authors.